# 'What happens when I can no longer care?' Informal carers' concerns about facing their own illness or death: a qualitative focus group study

Nan Greenwood,[1] Carole Pound,[2] Sally Brearley[1]

¹Faculty of Health, Social Care and Education, Kingston University and St George's University of London, London, UK
²Faculty of Health and Social Sciences, Bournemouth University, Poole, UK

**Correspondence to**
Dr Nan Greenwood;
Nan.Greenwood@sgul.kingston.ac.uk

## ABSTRACT

**Objectives** Older informal carers play an increasingly important role in supporting others with long-term health conditions. This study aimed to explore in depth the perspectives of older carers (70+ years) supporting others with a variety of conditions and disabilities focusing on their thoughts and experiences about when they are unable to continue caring.

**Design** Qualitative with four focus groups.

**Setting** Greater London, UK.

**Participants** 28 older carers (70+ years) recruited from the voluntary sector participated in this study. Most were women and many were spouses caring for partners with age-related conditions such as dementia, arthritis and visual impairment. Nearly a third were parents of adult children with severe physical or cognitive disabilities.

**Findings** Thematic analysis identified two main aspects for carers when contemplating the future—when they are unable to care in the short term or long term if they die or can no longer manage. Themes included the following: the impact of age, health conditions and relationships on future planning; anxiety about future care; carers' ambivalence and challenges in broaching the subject; interventions that might help older carers talk about and plan for the future of those they care for.

**Conclusions** Services need to be open to talking about this difficult topic. Our findings suggest that frank discussions about when older carers cannot care and having plans in place, whether these are financial or address other practical issues, makes it easier for all concerned. However, this issue is not easily broached and its timing and ways to access this support must be carefully and individually gauged. Future research with more diverse demographic groups is needed to improve understanding of these carers' perspectives. Research is also needed to develop interventions to support older carers to talk about and plan for the future.

## INTRODUCTION

With ageing populations, older informal, unpaid carers are growing in number and play an increasingly vital role in supporting others with long-term health conditions. For example, in the UK, numbers of adult carers in general are growing (up approximately 11% since 2001) but numbers of older carers

### Strengths and limitations of this study

► Planning for when they cannot care was highlighted by older carers themselves as a research priority, demonstrating the importance of the topic.

► Focus groups allowed in-depth exploration of older carers' experiences among peers.

► The older carer participants were recruited from the voluntary sector which may have influenced the findings and therefore future research should consider, for example, recruitment from primary care.

► Despite active attempts to recruit a diverse range of carer participants, we were unable to recruit many older male carers or those from minority ethnic groups.

► Employing focus groups for data collection meant that housebound carers and those unwilling to talk in groups about this sensitive topic were unable to participate and future research could include, for example, one-to-one interviews to allow their participation.

(65+ years) are increasing over three times more rapidly (by 35%). This rise is particularly marked for the oldest age group of carers aged 85 years or more who have grown in number by 128% over the last decade.[1]

The challenges of the caring role for adult informal carers in general have been well covered in the literature but although older carers are frequently included in studies exploring their experiences, the focus is seldom on them as a specific group.[2 3] This is despite the fact that they are often co-resident, care for longer hours and provide more intimate care than other age groups.[4] They might also be expected to find caring more challenging due to their age or because of the particular needs of those they care for as many are likely to have long-term disabilities and conditions such as dementia.

Recent research[5 6] with older carers has highlighted considerable anxiety about who will care for those they support if they

are unable to continue caring through illness or death. The same research identified this topic as a research priority both from the perspectives of carers themselves and professionals and volunteers supporting them. This concern applied to both carers of middle-aged and older adult children with disabilities and those supporting similar age spouses with conditions such as dementia or severe mental illnesses.[5 6] Previous research[7–12] looking into this issue has tended to focus on carers of adults with learning disabilities and often includes carers who are parents or siblings. These studies generally report anxiety about the future but despite this, there is frequently a lack of planning and reluctance on the part of informal carers to make plans. However, there is a dearth of research focusing specifically on the perspectives of older carers in general as opposed to carers of adults with learning disabilities.

## Aim

The aim of this study was therefore to explore in depth the perspectives of older carers (70+years) supporting people with a variety of conditions and disabilities focusing on their thoughts and experiences about when they are unable to continue caring through death or ill-health.

## METHODS

The primary focus of the project was identified in our earlier research[5 6] with older carers and was based on their experiences. These carers identified the topic as a research priority. The research was supported by an advisory group which included older carers, volunteers and professionals working with older carers. Two group members had dual roles as carers and professionals. This group encouraged the focus on the topic and provided feedback on our methods and approach.

## Participants

Carer participants were recruited by a third sector organisation based in Greater London, UK which supports informal carers. Recruitment was purposive with the intention of including a diverse range of older carers in terms of caring relationships (eg, caring for spouses or adult children with disability), length caring and ethnicity. To participate, carers had to be aged 70 years or older, be current or recent carers (in the last 2 years) and be able to participate fully in focus group discussions. Seventy years was selected as the minimum age, as increasingly people are working beyond the traditional retirement age of 65 years.

Recruiters, who were employed by the third sector organisation and who were familiar with individual carers and had worked with them in groups previously, contacted carers fitting the inclusion criteria by telephone, email or a mixture of both and described the project's aims and methods. Potential participants were forewarned that the groups would include discussions about potentially

sensitive or emotive topics. If they were happy with this, they were then invited to attend a focus group.

Given the dearth of research here, a qualitative, exploratory in-depth approach[13] using focus groups was selected. Although individual interviews have their advantages, focus groups were used as they allow participants to talk in depth and offer opportunities for discussion among peers with whom they share a common frame of reference.[14] Previous research with older carers has demonstrated the value of this approach for producing rich data.[15]

## Data collection

Groups were held at the end of 2018 and were digitally recorded and professionally transcribed. The groups were undertaken by researchers experienced in facilitating focus groups with older people and covering sensitive topics. All facilitators were women, middle-aged or older and one had been a carer herself. After giving written consent, participants provided some written background demographic information including, for example, their age and their relationships with care recipients.

Facilitators used a brief topic guide to ensure that important, relevant areas were covered. The primary focus throughout was on carers' thoughts about the future when they could no longer care. Topics included, for example, whether they had talked to anyone about it or made any plans and if so, how had they found the process. They were also encouraged to describe anything they had done or any support they had received that had helped with thinking about the future. A professional from the recruiting carer organisation was on hand throughout data collection in case any carers needed support.

Transcripts were anonymised immediately after transcriptions were completed. Recordings were destroyed once the transcriptions had been checked.

## Patient and public involvement

The research topic was identified by older carers based on their experiences and chosen as a priority for investigation in our previous work.[5 6] There were no patients involved in the study but the advisory group, which included older carers, professionals and volunteers contributed to the design of the study and oversaw it. They were not involved in recruitment. Once the study was completed, a lay summary was sent out to participants and a dissemination event for participants and other older carers was also held.

## Data analysis

Analysis was thematic[16] and started during data collection. Two authors (NG and CP) read and re-read the transcripts ensuring immersion in the data. Analysis started with independent open coding followed by discussion about the initial codes identified and emergent themes. Final themes were agreed on by the authors. Throughout the researchers strove to identify

and discuss any preconceptions, they may have had about the data or any personal experiences that might influence their analysis.

## FINDINGS

In all, 28 carers with an average age of just over 73 years participated in four focus groups (group size averaged seven but groups ranged from five to eight participants). All but one was currently caring. The single former carer had recently ceased being a carer but had cared for two decades previously. Three in five (60.7%) were supporting spouses or partners and approximately one-third (32.1%) were caring for adult children with long-term health conditions. There were more female (87%) than male carers and approximately two-thirds (67.9%) were White British. The next largest group described themselves as Asian Indian (21.4%). Most had been caring for a long time—the average length of caring was 18.5 years but six had been caring for three decades or more. Care recipients' primary health conditions included dementia, learning and physical disabilities from birth and a range of age-related disabilities such as visual impairment and arthritis. Further details of the carers' demographic characteristics are shown in table 1.

Focus groups lasted between 80 and 100 minutes and were lively with both very serious and light-hearted moments. A great deal of practical information and other support was shared among the carer participants both during and after the groups and many commented on how useful and beneficial it had been to share their thoughts and experiences with others in similar situations. However, despite being aware of the primary topic for the focus groups, there sometimes appeared to be reluctance among carers to discuss it with the carers often appearing to prefer to talk instead about, for example, inadequate services.

### Themes relating to planning for when carers can no longer care

Overall, it was clear that this is a very important topic which has considerable impact on older carers irrespective of whether they are caring for an adult child or a spouse of a similar age. Carers' discussions were often wide ranging but the themes described here relate specifically to their discussions around thinking about and planning for when they cannot care either in the short term due to a short illness, or long term if they can no longer manage or die . Themes included the following: the impact of diverse care recipients on planning for the future; future care of care recipients; the effect on carers of worrying about future care and care recipients' concerns; the relief planning can bring; carers' ambivalence and difficulties in broaching the subject with others; interventions or support that might help them with this issue.

**Table 1** Carer participant demographic characteristics

| n=28 (%) | |
|---|---|
| **Gender** | |
| Female | 24 (85.7) |
| Male | 4 (14.3) |
| **Age (years)** | |
| Median | 73.5 |
| Range | 70–87 |
| **Length caring (years)** | |
| Mean | 18.5 |
| Range | 1–40 |
| **Relationship to care recipient** | |
| Spouse/partner | 17 (60.7) |
| Parent | 9 (32.1) |
| Sibling | 2 (7.1) |
| **Ethnic group** | |
| White British | 19 (67.9) |
| Asian Indian | 6 (21.4) |
| Chinese | 2 (7.1) |
| Irish | 1 (3.6) |
| **First language** | |
| English | 21 (75.0) |
| Gujarati | 2 (7.1) |
| Chinese | 2 (7.1) |
| Other | 3 (10.7) |
| **Caring status** | |
| Current | 27 (96.4) |
| Former | 1 (3.6) |
| **Health conditions of care recipients** | |
| Dementia | 6 (21.4) |
| Visual impairment | 4 (14.3) |
| Arthritis/mobility issues | 4 (14.3) |
| Congenital disabilities | 4 (14.3) |
| Psychotic illnesses | 3 (10.7) |
| Diabetes | 2 (7.1) |
| Learning disabilities | 2 (7.1) |
| Other | 3 (10.7) |

Many care recipients had multiple health conditions. Only the primary ones as described by the carers are listed in the Table above. Where figures do not add up to 100% this is due to rounding of the figures.

Themes are listed as A–F and are described in detail below with their subthemes. All quotes are anonymised with pseudonyms. FG1–FG4 refer to the focus groups.

### A. The impact of age, health condition and relationship on discussing the future

In describing their individual situations, carers emphasised the centrality of the care recipients and their caring

relationships which they believed had a direct impact on how they thought or talked with others about the future. For example, parents of adult children with disabilities saw this as a more immediate issue as, being older, they were generally more likely to die before their child. They also thought that being a parent sometimes made it harder because they did not want to treat them as if they were still children. Significantly, though their thoughts were influenced by, for example, the care recipients' disabilities which affected how easy or difficult it would be to talk with them and how much they would understand or could be involved in discussions. For example, carers highlighted the differences between discussing the future with those who had severe cognitive and communication difficulties compared with people with physical disabilities who could fully understand what was being discussed. The following carer was afraid that, as his mother, she ran the risk of treating him as a child even though he was an intelligent adult.

> I can't treat him like a child, he's perfectly, I mean he's been to university, he's perfectly intelligent, but the problem is I'm his Mum so it's much harder for me to keep on at him, if he was ten, you know, it's easy, but he's 37. Frances FG2

In contrast, the wife of an elderly man with dementia who had been married for many years, wished they had discussed his wishes before his cognitive abilities had deteriorated. If they had done this and if she died first, she could be confident she had planned for his care as he would have wanted.

> I couldn't talk to him because he doesn't understand anything and he can't bear me talking, you know… yeah, in a way we're having the conversation but bless him, I wish I'd had it with him because I know specifically what I want but for me, but I can't have it [the conversation] with him. Alison FG4

This contrasted with other conditions, such as severe mental illness and others where care recipients could potentially engage in such conversations but they were regarded as very fragile, making carers extremely wary of raising the topic.

> I mean he is quite willing to talk about these things… Now I think one of the great things about not revisiting the subject too often is not wanting to put stress on him because if we do, it could lead to seizures. Claire FG3

This theme highlights the uniqueness of each caring situation and how this diversity influences thinking about the future.

### B.Short-term and long-term care for those they support

### Worrying about short-term back-up support if they are suddenly taken ill

Carers emphasised that there were two aspects to their concerns for the future. The first related to the short-term,

for example, if they suddenly became ill and unable to care with insufficient time to make plans. This carer who is in a reciprocal caring relationship with his wife said:

> … we have a huge problem of not only the future, what happens even in the present. If one of us is not well, not able to cope, then the other can't cope anyway. Manjeet FG1

Several carers had previously needed unexpected short-term support. This highlighted how lucky they had been and how the situation might have been considerably worse, making them realise they needed to think ahead. This carer who was supporting her husband with severe mental illness explained:

> I had an infection which… turned to septicaemia and I had to phone (my husband was in cloud cuckoo land, couldn't ring 111). I had to ring 111 myself and get the medics out and I was taken in for five days into hospital and had to leave my husband. He was actually crying on the doorstep… it was 3 o'clock on a Sunday afternoon. Where did I go to get help? Nowhere. I faced the grim reaper really, could have been fatal for me and I wasn't prepared for it. Margaret FG4

This carer who supports her husband with advanced dementia, faced similar concerns.

> I fell in the garden, split my head open… you know, I'm pouring with blood and my husband is just standing there. Lisa FG4

One carer suggested that as this was a pressing issue common to them all.

> I did wonder if the common theme here is an emergency - the weekend, the middle of the night? If, as a group of carers, we were to compile a letter and send it to all our GPs saying "This is our biggest fear, what can you come up with as the answer?" [lots of agreement] Michael FG4

These carers' discussions also clearly demonstrated their day-to-day anxiety about the care recipient should something happen to them.

### Worrying about the long-term care for those they support when they cannot care through death or ill health
Another significant and nagging worry was what would happen if the carers became physically unable to continue caring or died before the care recipient. This applied to all carers to a greater or lesser extent but was particularly obvious for the carers of adult children with disabilities.

> My main worry is his father's dead. He's an only child, and you know, what's going to happen when I die? [lots of agreement] Frances FG2

> It's pretty sure he'll probably survive me so that is the other leg of the problem. As well as searching for company for him, he obviously needs a lot of support on the business side of things. Tom FG4

Similarly, other carers supporting spouses of similar ages as themselves talked a great deal about this, although some appeared to have thought about it less, or to be slightly more fatalistic about it.

> I was 80 this year… you know, I haven't got years ahead, I don't know what's going to happen to my husband not **if** I die, **when** I die. I just hope he goes before I do. Margaret FG4

> … does somebody come in and take over, do they shove them in a home, or what? Elizabeth FG2

It was very apparent that these concerns about the long-term care of those they support were never far from carers' minds, irrespective of who they were caring for.

### C.The effect on carers of worrying about future care and care recipients' concerns
#### Anxiety about the future and having no plans in place
There was clear evidence of the anxiety and stress that not having made plans caused to carers. There was general agreement that this led to 'sleepless nights' but some took a fatalistic view.

> I'm desperately worried about the future and the only way I console myself is when I die, I won't know what happens! Which is a terrible way to think actually. [Laughs] Helen FG3

Despite trying to help their son to live independently, one couple had experienced many barriers because of the rarity of his illness.

> I've been concerned for years about what happens in the future because we he's living with us, he's a very difficult person to move on … he doesn't fall into the right categories to kind of go into other accommodation. Claire FG3

Carers emphasised that care recipients were also very afraid of what the future might bring, even if they were often unwilling to talk in depth about it. The following carer was supporting her spouse.

> He is scared, he is very scared that I'm going to die first … He said "I don't know what I'd do if you died first". He said "I'll panic." It's somewhere where he doesn't want to go, he can't envisage this. Margaret FG4

#### Financial anxiety
Financial concerns were a significant part of carers' worries. They were often vaguely aware of nursing home care costs but felt they were probably beyond their reach or unsustainable long term.

One carer had devoted a considerable amount of time to trying to sort out her son's financial situation, so he could remain in their home when she died. His limited benefits meant that he would be unable to cover household bills. As a result, although she wanted him to inherit the house, he would have to move out after her death.

The continuously changing landscape in terms of legislation and benefits made planning even harder.

> Can he afford to keep the house? … he lives with me, but can he manage to pay for all the bills in the house with £325 every four weeks? … I'm trying to update my will now but I don't have a crystal ball, I don't know what the rules will be then… They're constantly changing. Deborah FG2

Having no plan for the future led to a great deal of anxiety for carers and their financial concerns tended to make this worse.

### D.Ambivalence and difficulties in broaching the subject with others
Although the topic clearly had a lot of resonance with carers, they emphasised the challenges of discussing it. This was partly due to their own ambivalence which they described in some detail but was also because other people including, families, care recipients and professionals, often appeared unwilling to talk about it. The only groups that generally appeared open to discussing it were lawyers and financial advisors.

### Carers' own ambivalence
Carers' own ambivalence meant they described finding excuses not to persevere when trying to talk with others or allowing themselves to avoid thinking about it when there was no acute problem. These carers felt they needed to be in the right frame of mind and to be feeling 'strong'.

> I pushed it to the back of my mind because he's physically very well and there's nothing wrong with me. Janet FG2

> We try to keep cheery and take every day as it comes, but it's just, you know, because it's only the two of us. Alison FG2

However, several carers had managed to talk to the care recipients and said that the anticipation of the conversation was actually worse than the conversation itself.

> It was the fear of it, I think, more than anything… For me I think the view of having the conversation, rather than the conversation itself, you know, [lots of agreement]. When I had one with my son, I prepared for a week … it was over in about two min, so that was easy. Deborah FG2

However, one carer who admitted she was not ready to make plans herself, said she felt under pressure to organise institutional care for her severely disabled son. Not only did she not want to stop caring but she also felt she would never really be sure that he was happy living away from her.

> Social Services said I must look round now because before I go… my son is autistic, he doesn't know, he can't speak, if he doesn't like the place he can't tell me, so it's difficult for me to find somewhere… so I want to keep him as long as possible if I keep myself

strong. I love him so much … so I feel "Oh so what if I let him go? I'm just nothing. I live for him. Jasmine FG3

However, when asked directly if they really did want to discuss future planning, most carers emphasised that they did.

### Families' and care recipients' ambivalence

Family members were often unwilling to have conversations about when the carer died or was unable to care apparently preferring to wait until a crisis when they would be forced to confront it. The following conversation took place in the fourth focus group.

I think my daughter would probably say, if she said anything, "Well I'll have to deal with that when it happens won't I? I'm not going to think about it now." Margaret FG4

It's like my son, he's in denial …You know, "It's never going to happen." Tom FG4

No. "You're going to live for ever and ever amen", yeah. Michael FG4

In another group, a carer remarked:

… when you start to talk about it, people say "Why are you being so morbid?" FG3

However, the carer participants also said they were unwilling to broach the subject with their adult children who might take over the care because they had their own responsibilities.

They are very, very busy … they are working from morning until night … you know, it's difficult, they've got their own children as well to look after. Aadi FG1

Care recipients sometimes made it difficult, appearing to want to delay the conversation.

Well I tried with my husband but as I say, he's so blooming awkward and cantankerous. [He says] "Don't worry about it, I'm not dead yet, and you're not dead yet." Elizabeth FG2

A carer who was gradually finding it harder to care for her physically disabled son wanted to be sure that he was settled and happy somewhere before she really was unable to look after him. However, she worried he would feel she was pushing him out of their home.

My difficulty is my son keeps saying "Oh but…" the minute we start to talk about the future and about me not going to be here for ever and he starts saying "But Mum can we not wait till that happens?"… So it's almost like [he thinks] I'm going to be pushing him out. Tuhina FG3

She continued saying she suspected that he had already reflected on the future but superstitiously did not want to think about it in case this somehow precipitated something happening to her.

So, there is an outside element that he may have already thought things through in his own mind but isn't willing, because I suspect my son's thought through in his own mind but if I speak about it, it might happen so "Don't let's speak about it." [General agreement] Tuhina FG3

In the fourth focus group, a carer supporting her husband with early dementia said she had tried to talk to him but had failed because he turned it into a joke refusing to engage in the conversation.

Other carer participants joined in the discussion agreeing with her.

I mean my husband, when you say that to him, [he says] "You'll have to kill me."… They can't face the thought of losing you, they can't. Janice FG4

That's right, yeah. Tom FG4

They won't go there. No, they won't go there. Sarah FG4

That's exactly, yes. Janice FG4

You don't want to upset him. Sarah FG4

Not only did the older carers find it difficult to talk to those close to them about the future but professionals were often unwilling to engage in these conversations as well.

### Statutory services' apparent ambivalence

Carers felt that health and social care professionals should proactively broach the subject but with the exception of the one carer described earlier, their experience suggested there was little appetite for this. Even when carers tried to initiate discussions, there was reluctance.

I've been to Social Worker, I've been to support worker as well and I asked them. They said "As long as you manage, carry on. If there is any problem in future come to us." Aadi FG1

This carer described what she thought was an ideal situation to have a conversation about what happened if she died and there was no one to support her husband with dementia but again, it was avoided.

When I said at the hospital… the consultant was talking about power of attorney and all these things, which was great, she was really lovely, and I said, "The only worry is if I go first" and she went "Heaven forbid!" Alison FG4

There was a sense among the participants that no one wanted to talk about it with them because they were currently managing.

I don't think the doctors are interested in what you do as long as you are there. Elizabeth FG2

Despite their anxieties and desire to talk to others, carers clearly faced considerable challenges in trying to

discuss their concerns with others but planning some-times helped them worry less.

### E.The relief planning can bring

It appeared that most carers had considered what might happen if they could not care but they varied hugely in how successfully, if at all, they had made plans. Several had started financial planning, helping to reduce their anxiety.

> When I die, I've got a house which obviously belongs to my three children but I would want something done and of course he (the lawyer) said "Oh you could set up a trust fund" and, so just that [helped bring] a little bit of peace. Caroline FG3

One approach was to investigate local care homes to find out if they would take someone at very short notice. This process had brought this carer some relief.

> If I went [first], they have emergency beds They don't officially do respite but they do keep emergency beds. So I would hope my children would be able to ring them and they would say "We can have him." Alison FG4

The following carer had tried unsuccessfully to broach the subject if either of them died with his wife. However, after researching he came up with a strategy that considerably helped start the conversation. This had then continued long after the initial discussions and was clearly a relief.

> The way I opened the conversation about end of life with my wife, I'd read it in some books… I said, "Listen, the important thing about your funeral is what hymns? What flowers do you want?" And all these things that were totally unimportant but it opened the conversation. … by bringing it in almost in a not too serious way, it got her thinking … but it all starts leading round to sort of opening the conversation, not being scared to talk about it… we still talk about it, probably every week we come up with something. Tom FG4

In describing their experiences, the carers here offered insights into how others might cope in their situations. This led to discussions about how services could provide them with support.

### F.Interventions or support that might help older carers

Many carers described not knowing how to start the process of planning and wanted help and information.

> Where do we start? Manjeet FG1

> If I popped me clogs, and you know, how do you go about it, what are the other options, somebody just visiting [him] now and again, or somebody living in? Janet FG2

There was generally consensus that the statutory sector was more focused on care recipients and lacked time and possibly the expertise for this. Generally, the voluntary sector was thought to be better placed.

I don't think Social Workers necessarily have that expertise because there's no point in opening up a hornet's nest and walking away from it and leaving somebody with all the questions. Louise FG3

Charities whose focus was the carers not only keep up to date with legislation but could provide skilled facilitators for discussion and, for example, experts in benefits or legislation.

> Everything they've done gets updated so that people can benefit from that expertise. Because it would certainly be a weight off my mind that somebody was saying "Hey, did you know this?" Theresa FG3

Many carers thought that support with future planning could be provided in groups, rather than individually.

> It might be less threatening as a group… with somebody really skilled who could initiate those conversations. Louise FG3

This carer suggested initially having two simultaneous groups—one for care recipients and another for carers—followed up by bringing the two groups together.

> Ideally you'd have two separate groups with different groups facilitating and hopefully then you could get them merging so that you can actually talk maybe with other people. So that, for example, I might be talking with Mary's son as part of the group and eventually Mary and her son will be talking. Theresa FG4

It was thought that care recipients would find it easier to appreciate the importance of talking about this from someone other than their own carer. Peer support was also thought to make it easier for everyone.

## DISCUSSION

This study highlights the anxiety felt by older carers supporting others with a range of health conditions and adds significantly to the literature about older carers' experiences and support needs. Research in the area is limited and this study considerably improves our understanding of the challenges of being an older carer. By focusing on this important group, who do so much to support others often at a physical and emotional cost to themselves,[4–6] we can begin to identify how we can best support them in their vital role. Worrying about the future care of their care recipients results in anxiety and stress for older carers. Those carers that had successfully spoken to others or made plans for this, emphasised the relief this had brought. Their concerns relate to both short-term plans for care should they suddenly become acutely ill but also if they are permanently unable to care through death, illness or because of their advanced age. Many carers highlighted that, given their age, they saw this likelihood as almost inevitable. These two aspects of their fears require different support. For many knowing there was somewhere that they could contact in an

emergency would have helped considerably but the need to have long-term plans, including financial plans in place was also very important. Overall, the voluntary sector was thought by these older carers to probably be best placed to support them with this planning.

The findings highlight both the importance for carers of being able to plan for the future care of those they support but also how difficult they found this. Not only are other people often unwilling to talk about it but factors such as who to involve in these discussions and not knowing how to go about putting plans in place make doing this hard. However, many carers admitted that they themselves were a barrier as they did not find it easy to broach the subject. It is essential that we understand these obstacles if we are to support carers and our findings add significantly to this. Previous research focussing on older carers of adult children with learning disabilities[7–12] identified similar concerns, barriers and lack of future planning among carers but did not investigate older carers' concerns for care recipients of a similar age to themselves. A recent focus group study of older carers of people with learning disabilities reported that few had made future plans and that they thought support with this was limited.[17] The authors concluded that proactive initiatives are needed to allow timely planning. We agree with these researchers but our findings suggest that these issues apply to older carers irrespective of who they are caring for.

Our focus groups were lively but the flow of the discussions appeared to reflect carers' thinking and talking about the topic in general. Conversations about when they could no longer care often started with carers talking about future planning but frequently drifted onto descriptions of the more immediate challenges of caring or the limited statutory support available. This may have been a reflection of the fact that, although carers said that they wanted and needed to talk about this, when put on the spot, they found it difficult. It was also striking that carers often joked about when they died, for example, referring to 'popping their clogs' or describing unusual, amusing plans for disposing of their ashes. This may have been one way of talking about this difficult topic and similar to the carer who described how he had successfully broached the subject initially in a light-hearted way. Another striking aspect of the groups was how often carers remarked on how beneficial and 'therapeutic' they had found the opportunity during the focus groups to talk to other carers with similar concerns and although support was on hand during and immediately after the focus groups, in the event, it was not requested by any carers even though they were reminded that it was available. This may endorse the belief held by many of our participants that a group intervention with peer support might be valuable. More needs to be known about interventions to facilitate initiating discussions and planning for future care but our findings suggest these may require creative, individualised approaches.

Importantly, this study was identified by older carers as a research priority[5 6] and our findings have provided in-depth insight into what is a major concern for older carers. Discussions among the carer participants and points during the focus groups when there was clear agreement between participants gave insights that may not have been achieved using other data collection methods. However, housebound carers and those unwilling to talk in groups about this sensitive topic could not participate and future research could include, for example, one-to-one interviews to allow their inclusion and to potentially add further depth to the data. Furthermore, although we included a variety of carers in terms of their caring relationships and length of caring, all were in contact with the voluntary sector which may have influenced some of their discussion, for example when suggesting that the voluntary sector was ideally placed to offer support. We were also unable to recruit many older male carers or those from minority ethnic groups despite active attempts to do so. The participants here were all recruited from the voluntary sector and future research should include carers recruited more widely, for example, from primary or secondary care to avoid this potential bias in the findings. Additionally, the carers were all fully aware of the focus of the research and possibly some carers, who did not consider the topic to be relevant or of interest of them, may have declined to participate. Finally, it is recognised that many people in caring roles are not identified by themselves or services as being carers and therefore to not receive support.[18] Efforts should be made to ensure this important group is included in future research.

Given the dearth of research here and the focus of previous research on older parental carers, for example,[7–12] it would be helpful for future research to focus on older carers supporting someone with health conditions such as dementia, as this would allow a more nuanced analysis of the impact of the care recipients' diagnosis. This approach may also help provide more individualised support for these carers. Further research with more diverse demographic groups both in terms of ethnicity and gender is needed to improve understanding of their perspectives. Research is also needed to develop interventions to support older carers to talk about and plan for the future.

## CONCLUSIONS

Services need to be open to talking about this difficult topic with carers and care recipients. Our findings suggest these discussions support carers to make financial and practical plans and that the process of talking can bring relief in itself. However, this is not an issue that is easily broached and the timing of and ways to access this support must be carefully and individually gauged.

**Acknowledgements** We are grateful to Wellcome Trust who funded the study, the voluntary sector organisations who recruited participants, the carer participants and the advisory group of carers and volunteers who oversaw the study.

**Contributors** NG led the study and drafted the paper. Data analysis was undertaken by NG and CP. All three authors (NG, SB and CP) took part in data collection and added to and approved the final version of the paper.

**Funding** This research was supported by the Wellcome Trust grant number [209343/Z/17/Z].

**Competing interests** None declared.

**Patient consent for publication** Not require.

**Ethics approval** Ethics approval was gained from the Faculty Research Ethics Committee, Kingston University, London (Ref: FREC 2017-11-004).

**Provenance and peer review** Not commissioned; externally peer reviewed.

**Data availability statement** No data are available.

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
