## [Reviewer comments · BMJ Open]

ARTICLE DETAILS

TITLE (PROVISIONAL)	'What happens when I can no longer care?' Informal carers' concerns about facing their own illness or death: a qualitative focus group study.
AUTHORS	Greenwood, Nan; Pound, Carole; Brearley, Sally

VERSION 1 – REVIEW

REVIEWER	Dr Warren Donnellan University of Liverpool, UK.
REVIEW RETURNED	16-Apr-2019

GENERAL COMMENTS	Thank you for the opportunity to review this paper. The paper focuses on an interesting and worthwhile area of research. Unfortunately it is not ready for publication in this journal. I outline the reasons why below under each section of the paper: Article summary • You say that it is strengths and limitations but the points you provide are more summary of findings. Points made should be consistent with the subtitle. Introduction • All bases covered but should provide a more in-depth, nuanced review of the relevant literature, particularly in the area of investigation (last paragraph). Methods • Use of focus group method is well justified with evidence.• Would benefit from subheadings for greater clarity: i.e. participants, data collection, data analysis.• How did you assess whether participants were able to participate fully in focus groups?• I would bring the last paragraph of the method sooner as it is important and a key strength of the work. Findings • Generally good use of quotes and insightful analysis.• First paragraph of findings and Table 1 belong in Participants section of Method; they are demographic characteristics rather than findings. Findings should start with 'A great deal of practical information...'• Unclear why mean and median provided for age (years); it is either/or• Rationale for using all but one current carers should be clearer.• Could you include a more informative heading than 'Themes'?
--

	 • Try not to end findings subsections on a quote. Similarly I think would benefit from a summary at end of findings which ties everything together, rather than ending on a quote. • The themes listed on p8 do not clearly describe how the A, B, and C themes relate to the subthemes within them in subsequent paragraphs – confusing for the reader, try to use the same wording. • Should include pseudonyms for individual participants within each focus group, rather than just the focus group number. You do this later on, but only when illustrating conversations between participants. • Avoid using ‘they’ when start new subsection, don’t assume the reader knows who they are. E.g. p10 • For A, the impact of age and relationship is explained in text but not really supported by quotes. Only the impact of condition is illustrated by quotes, p9 • Make it clearer in the findings that the theme of interventions or support came spontaneously without asking, this is very interesting. Discussion  • Should include more discussion of findings in relation to extant literature, specifically comparison of experiences of different carer subgroups. A more detailed literature review would have helped here. • Clearly separate out study evaluation, future research ideas from discussion of findings in relation to extant literature. • Expand on applied relevance of findings. • Whilst it can be good to include a varied sample of carers, it does dilute the findings somewhat. A key area of future research would be standardising care recipient condition to provide a more nuanced account of impact of condition on discussing the future, e.g. dementia carers.
--	---

REVIEWER	Morag Farquhar University of East Anglia, UK
REVIEW RETURNED	03-May-2019

GENERAL COMMENTS	Manuscript ID bmjopen-2019-030590: "What happens when I can no longer care?' Informal carers' concerns about facing their own illness or death: a qualitative focus group study. Important topic which this papers begins to explore. Paper would benefit from consideration of the following points:  1) Strengths and limitations bullet points – need to include in the bullet point about future research that recruitment should consider via primary care settings rather than via voluntary sector groups. 2) Introduction, para 2, last sentence: could add something about “or the particular caring needs of those they care for” (as care recipients are more likely to ne those with long-term conditions, long-term disabilities or dementia. 3) Methods, para 1: main concern is the bias inherent in the sampling frame (third sector organisation). 4) Methods, para 2: disagree that focus groups give in-depth data, especially with this group size – may remove “in-depth” from first sentence. 5) Methods, para 3: who did the recruiting (what was their usual role?) and how were participants contacted (post, email. Phone, face to face, poster)? Very light on details of recruitment.
--

	6) Methods, para 4: information about transcription process and recordings destruction is in the wrong place – should come later, after the data collection description and just before the analysis. 7) Methods, para 5: good that a support mechanism was in place for carer participants at the focus groups but was it needed? If so, how many. Useful information for other researchers in study planning. 8) Findings, para 1: how many participants per group? Average is n=7 (quite big for a sensitive topic), were some bigger? 9) Findings, para 2. Last sentence: wonder if some of the reluctance to discuss the topic was due to a focus group method – might they have spoken more openly about it in one-to-one interviews? Needs reflecting on. Also sentence needs re-working as is a difficult read as presently constructed. 10) Findings, bottom of page 9: may be better to describe this carer as being “in a reciprocal caring relationship”? 11) Findings, top of page 10: “unexpected short-term BACK-UP support”? 12) Findings, top of page 10: something wrong with first sentence – does situation need pluralising maybe? Sentence needs reworking. 13) Findings: some intro text to quotes is in past tense, some in present – needs to be in past e.g. “This carer supports...” should be “A carer who supported...” 14) Findings, quotes: is it possible to indicate which participant within a focus group gave the quote as at present all we can judge is whether the quotes come from a range of FGs, not people. 15) Findings: advise removing the term “loved ones” throughout – care recipients are not always the “loved ones” of carers; relationships can be very strained. 16) Findings, section on “Statutory services’ apparent ambivalence”: this section contradicts the quote at bottom of page 12 where the carer says social services were trying to get her to consider this, but she was reluctant. 17) Findings, quotes: full stops missing within some quotes. 18) Findings, quote on page 15 about end of life planning: not clear if the is about end of life planning for the care-recipient or the carer. 19) Discussion, first sentence: needs re-working as the study doesn’t highlight the impact of anxieties, it highlights the anxieties. 20) Discussion, second sentence: not convinced the data says that worrying about this resulted in “considerable anxiety and stress”. It was certainly a source of anxiety. 21) Discussion, top of page 17: the voluntary sector was maybe considered a best placed to support these carers as they were recruited from that setting. Bias? 22) Discussion, page 17, first main para: delete “amongst carers supporting”? 23) Discussion: you might like to look at the Carer Support Needs Assessment Tool (CSNAT) Approach at http://csnat.org/ - the CSNAT includes an item about “knowing what to expect in the future” which could help start a supportive conversation with a health or social care professional about this topic. 24) Final thought that could be added to the discussion: support from professionals with these sorts of concerns can only happen if carers are visible and identified in the first place. We need to know who carers are. The very fact that you had to recruit through a voluntary agency is telling,
--	--

VERSION 1 – AUTHOR RESPONSE

Reviewers' Comments to Author

Thank you for these helpful, constructive comments.

Reviewer: 1

Please leave your comments for the authors below

Thank you for the opportunity to review this paper. The paper focuses on an interesting and worthwhile area of research. Unfortunately it is not ready for publication in this journal. I outline the reasons why below under each section of the paper:

Article summary

- You say that it is strengths and limitations but the points you provide are more summary of findings. Points made should be consistent with the subtitle.

Thank you for this comment. It was also highlighted by the editor and has been amended (p4).

Introduction

- All bases covered but should provide a more in-depth, nuanced review of the relevant literature, particularly in the area of investigation (last paragraph). *Thank you for highlighting this. The last paragraph has been revised to highlight the dearth of research focussing on older carers themselves (p5). It now reads:*

Recent research[5,6] with older carers has highlighted considerable anxiety about who will care for those they support if they are unable to continue caring through illness or death. The same research identified this topic as a research priority both from the perspectives of carers themselves and professionals and volunteers supporting them. This concern applied to both carers of middle-aged and older adult children with disabilities and those supporting similar age spouses with conditions such as dementia or severe mental illness[5,6]. Previous research[7-12] looking into this issue has tended to focus on carers of adults with learning disabilities and often includes carers who are parents or siblings. These studies generally report anxiety about the future but despite this, frequently a lack of planning and reluctance on the part of informal carers to make plans. However, there is a dearth of research focusing specifically on the perspectives of older carers in general as opposed to carers of adults with learning disabilities.

Methods

- Use of focus group method is well justified with evidence.

Thank you for this positive comment.

- Would benefit from subheadings for greater clarity: i.e. participants, data collection, data analysis.

These have been added (p5,6)

- How did you assess whether participants were able to participate fully in focus groups?

Thank you for highlighting this. Another reviewer commented on this and so the text has been changed to reflect both reviewers' comments. Recruitment was undertaken by two voluntary sector workers who were familiar with the carers and who were ideally placed to know if they could participate fully. They took considerable care to ensure that all potential participants were aware what the focus groups entailed. We have added the following to the paper (p6) to clarify this: 'Recruiters who were employed by the third sector organisation and who were familiar with individual carers and had worked with them in groups previously contacted carers fitting the inclusion criteria by telephone, email or a mixture of both and described the project's aims and methods.'

- I would bring the last paragraph of the method sooner as it is important and a key strength of the work.

Thank you – this has been done.

Findings

- Generally good use of quotes and insightful analysis.
- First paragraph of findings and Table 1 belong in Participants section of Method; they are demographic characteristics rather than findings. Findings should start with 'A great deal of practical information...'

Thank you for this. We have always included participant demographic characteristics as part of the findings in previous BMJ Open papers and publications and have therefore not moved the participant

demographics.

- Unclear why mean and median provided for age (years); it is either/or
We included both the mean and median ages for clarity (as the mean can be skewed by outliers). However, as they are very similar, we have removed the mean age (p8).

- Rationale for using all but one current carers should be clearer.

Unfortunately, this was how recruitment went – we were looking for both current carers and people who had recently been caring but the majority of carers engaged in the voluntary sector are current carers. However, the former carer in question was only recently a carer.

The following (p8) has been added: The one former carer had recently ceased being a carer but had cared for two decades previously.

- Could you include a more informative heading than 'Themes'? *This has been expanded to read (p8):*

Themes relating to planning for when carers can no longer care

- Try not to end findings subsections on a quote. Similarly I think would benefit from a summary at end of findings which ties everything together, rather than ending on a quote.

Thank you for this suggestion. We have gone through and added some linking and concluding statements (p10-17)

- The themes listed on p8 do not clearly describe how the A, B, and C themes relate to the subthemes within them in subsequent paragraphs – confusing for the reader, try to use the same wording.

Thank you for this highlighting this. We have now explained this on p 9.

- Should include pseudonyms for individual participants within each focus group, rather than just the focus group number. You do this later on, but only when illustrating conversations between participants.

The rationale for only assigning the focus group number was to reduce any chance of carers being identified (although we had already anonymised quotes, we wanted to be doubly sure of this). However, we have gone back and looked at the quotes and concluded that they cannot be identified so individual carers have now been identified with pseudonyms.

- Avoid using 'they' when start new subsection, don't assume the reader knows who they are. E.g. p10. *Thank you. We have added carers instead of 'they' e.g. on p 11.*

- For A, the impact of age and relationship is explained in text but not really supported by quotes. Only the impact of condition is illustrated by quotes, p9. *Thank you for pointing this out. We thought this was clear but have added the following to clarify it (p9):* This carer was afraid that as his mother, she ran the risk of treating him as a child even though he was an intelligent adult.....

In contrast, this wife of an elderly man with dementia who had been married for many years, wished that they had discussed his wishes before his cognitive abilities had deteriorated so that if she died first, she could be confident she had planned for his care as he would have wanted.

- Make it clearer in the findings that the theme of interventions or support came spontaneously without asking, this is very interesting. *Thank you. We have added a sentence (p 7) when describing the methods to clarify this as we encouraged carers to talk about anything that helped them with this issue. This naturally led onto what help they would like.:* They were also encouraged to describe anything they had done or any support they had received that had helped with thinking about the future.

Discussion

- Should include more discussion of findings in relation to extant literature, specifically comparison of experiences of different carer subgroups. A more detailed literature review would have helped here.

Thank you for this comment. As we explained, there is very little literature on this topic (except for parental carers of adult children with disabilities) and what we identified as most relevant is already mentioned in the Introduction.

- Clearly separate out study evaluation, future research ideas from discussion of findings in relation to extant literature.

Thank you for this. We have revisited the Discussion and feel that overall our approach basically worked primarily because of the dearth of other research focusing on older carers as opposed to carers of specific conditions. However, we have reorganised the Discussion in places (p17-19) to make it clearer where our research fits in with the findings of other authors and where our study could have been improved.

- Expand on applied relevance of findings. *Thank you for this helpful comment. We have added (p17):* Research in the area is limited and this study considerably improves our understanding of the challenges of being an older carer. By focusing on this important group who do so much to support others often at a physical and emotional cost to themselves[4-6], we can begin to identify how we can best support them in their vital role.
- Whilst it can be good to include a varied sample of carers, it does dilute the findings somewhat. A key area of future research would be standardising care recipient condition to provide a more nuanced account of impact of condition on discussing the future, e.g. dementia carers.

Thank you for this helpful comment. We have added the following (p19) to highlight this. Furthermore, it might be helpful for future research to focus on older carers supporting someone with similar health conditions, such as dementia, as this would allow a more nuanced analysis of the impact of the care recipients' diagnosis. This approach may also help provide more individualised support for these carers.

Reviewer 2

Important topic which this papers begins to explore. Paper would benefit from consideration of the following points:

1) Strengths and limitations bullet points – need to include in the bullet point about future research that recruitment should consider via primary care settings rather than via voluntary sector groups. *Thank you for this helpful comment. We have added this to the bullet points (p4) to read:* The older carer participants were recruited from the voluntary sector which may have influenced the findings and therefore future research should consider, for example, recruitment from primary care.

2) Introduction, para 2, last sentence: could add something about “or the particular caring needs of those they care for” (as care recipients are more likely to be those with long-term conditions, long-term disabilities or dementia. *Thank you – that is a really useful addition. We have added (p5) ...age or because of the particular needs of those they care for as many are likely to have long-term disabilities and conditions such as dementia.*

3) Methods, para 1: main concern is the bias inherent in the sampling frame (third sector organisation). *Thank you for drawing this to our attention. We have highlighted this as a bullet point in the Limitations section see above (p4) and similarly in the text on p 19.*

The participants here were all recruited from the voluntary sector and future research should include carers recruited more widely, for example, from primary or secondary care to avoid this potential bias in the findings.

4) Methods, para 2: disagree that focus groups give in-depth data, especially with this group size – may remove “in-depth” from first sentence.

Thank you for this comment. However, we were involved in both facilitating and analysing the focus group transcripts and we believe that the data were in-depth and have therefore retained this word. However, we have also added later that further research should include one-to-one interviews (p19).

5) Methods, para 3: who did the recruiting (what was their usual role?) and how were participants contacted (post, email, phone, face to face, poster)? Very light on details of recruitment. *This was an important omission, thank you for highlighting it. The other reviewer also commented on this and we have revised the paper according to both your comments. It now reads (p6):*

'Recruiters who were employed by the third sector organisation and who were familiar with individual carers and had undertaken work with them in groups previously contacted carers fitting the inclusion criteria by telephone, email or a mixture of both and described the project's aims and methods.'

6) Methods, para 4: information about transcription process and recordings destruction is in the wrong place – should come later, after the data collection description and just before the analysis. *Thank you. This section has been moved to p7 as suggested.*

7) Methods, para 5: good that a support mechanism was in place for carer participants at the focus groups but was it needed? If so, how many. Useful information for other researchers in study planning. *As it turned out this support was not needed. Indeed, as we mentioned, many carers said they had found the discussions in the groups beneficial. However, we have now said on p 18 that no support was needed.* ‘Another striking aspect of the groups was how often carers remarked on how beneficial and ‘therapeutic’ they had found the opportunity to talk to other carers with similar concerns and although support was on hand during and immediately after the focus groups, in the event, it was not requested by any carers even though they were reminded that it was available.’

8) Findings, para 1: how many participants per group? Average is n=7 (quite big for a sensitive topic), were some bigger? *We intended to have between 5 and 7 participants per group. Past experience suggested that we need to over-recruit as carers often cannot turn up because of their caring role. However, in all cases except one, all the participants managed to attend. This meant two groups had 7 participants, and one each 5 and 8. We have (p7) added: Twenty-eight carers with an average age of just over 73 years participated in four focus groups (group size averaged seven but groups ranged from five to eight participants).*

9) Findings, para 2. Last sentence: wonder if some of the reluctance to discuss the topic was due to a focus group method – might they have spoken more openly about it in one-to-one interviews? Needs reflecting on. *Thank you for this comment. We had already mentioned that future research should include on-to-one interviews (p18). It reads: However, housebound carers and those unwilling to talk in groups about this sensitive topic could not participate and future research could include, for example, one-to-one interviews to allow their inclusion.*

Also sentence needs re-working as is a difficult read as presently constructed. *Thank you. We have reworded the sentence to read (p7/8): However, despite being aware of the primary topic for the focus groups, there sometimes appeared to be reluctance amongst carers to discuss it with the carers often appearing to prefer to talk instead about, for example, inadequate services.*

10) Findings, bottom of page 9: may be better to describe this carer as being “in a reciprocal caring relationship”? *This is a very good suggestion, thank you. We have changed it (p10) to read: This carer who is in a reciprocal caring relationship with his wife said...*

11) Findings, top of page 10: “unexpected short-term BACK-UP support”? *Thank you for this. It now reads: **Worrying about short-term back-up support if they are suddenly taken ill***

12) Findings, top of page 10: something wrong with first sentence – does situation need pluralising maybe? Sentence needs reworking. *Thank you for highlighting this. It now (p10) reads: Several carers had previously experienced the need for unexpected short-term support. This highlighted how luck had been on their side and how the situation might have been considerably worse, making them realise they needed to think ahead. This carer who supported her husband with severe mental illness explained:...*

13) Findings: some intro text to quotes is in past tense, some in present – needs to be in past e.g. “This carer supports...” should be “A carer who supported...” *We have gone through and changed this.*

14) Findings, quotes: is it possible to indicate which participant within a focus group gave the quote as at present all we can judge is whether the quotes come from a range of FGs, not people. *Thank you. We have gone through and identified the individuals with pseudonyms to make this clearer.*

15) Findings: advise removing the term “loved ones” throughout – care recipients are not always the “loved ones” of carers; relationships can be very strained. *Thank you for this suggestion – we were also slightly ambivalent about this phrase and have changed it on the following pages 3,5,10,11,13,17..*

16) Findings, section on “Statutory services’ apparent ambivalence”: this section contradicts the quote at bottom of page 12 where the carer says social services were trying to get her to consider this, but she was reluctant.

Thank you for highlighting this. The carer who mentioned that social services wanted to think about and plan the future care of her son was unusual and we did not make this sufficiently clear in the paper. We have added the following (p15): Carers felt that health and social care professionals should proactively broach the subject but with the exception of one carer described earlier, their experience suggested there was little appetite for this. Even when carers tried to initiate discussions, there was reluctance.

17) Findings, quotes: full stops missing within some quotes. *We have checked and corrected this.*

18) Findings, quote on page 15 about end of life planning: not clear if the is about end of life planning for the care-recipient or the carer. *Thank you – we have identified two quotes where this might be unclear and have clarified them both. We have clarified both (p 15 and 16)*

19) Discussion, first sentence: needs re-working as the study doesn’t highlight the impact of anxieties, it highlights the anxieties.

This has been amended (p17) to read: This study highlights the anxiety felt by older carers supporting others with a range of health conditions and adds significantly to the literature about older carers’ experiences and support needs.

20) Discussion, second sentence: not convinced the data says that worrying about this resulted in “considerable anxiety and stress”. It was certainly a source of anxiety. *Thank you. We believe our data does demonstrate that thinking and worrying about the future did cause stress and anxiety but have moderated the sentence by removing the word ‘considerable’.*

21) Discussion, top of page 17: the voluntary sector was maybe considered a best placed to support these carers as they were recruited from that setting. Bias? *Thank you for highlighting this potential source of bias – it is a very good point. We have changed the sentence on p18 slightly to read: Overall, the voluntary sector was thought by these older carers to probably be best placed to support them with this planning.*

We have also highlighted our recruitment from the voluntary sector as a potential source of bias on p19 to read: Furthermore, although we included a variety of carers in terms of their caring relationships and length of caring, all were in contact with the voluntary sector which may have influenced some of their discussion, for example when suggesting that the voluntary sector was ideally placed to offer support.

22) Discussion, page 17, first main para: delete “amongst carers supporting”? *Thank you for highlighting this error. It has been corrected.*

23) Discussion: you might like to look at the Carer Support Needs Assessment Tool (CSNAT) Approach at <http://csnat.org/> - the CSNAT includes an item about “knowing what to expect in the future” which could help start a supportive conversation with a health or social care professional about this topic. *Thank you for this suggestion. We had looked at it previously and although this is a great tool, we do not think it really addresses the carers specific concerns about care of their care recipient if they are not able to do so. We have therefore not cited it here.*

24) Final thought that could be added to the discussion: support from professionals with these sorts of concerns can only happen if carers are visible and identified in the first place. We need to know who carers are. The very fact that you had to recruit through a voluntary agency is telling,

Thank you. In fact, we chose to recruit through the voluntary sector for several reasons (e.g. we wanted to run focus groups and wanted to undertake them in a venue familiar to the carers) although we accept this had potential bias. However, the point about so many carers not being visible (or indeed even seeing themselves as carers) is an extremely important one. We have therefore added (p19): Finally, it is recognised that many people in caring roles are not identified by themselves or services as being carers and therefore to not receive support[18]. Efforts should be made to ensure this important group are included in future research.

VERSION 2 – REVIEW

REVIEWER	Dr Morag Farquhar University of East Anglia (UEA), UK
REVIEW RETURNED	17-Jul-2019

GENERAL COMMENTS	This is a much improved manuscript. The only point I would re-raise is my Point 23... I NOTED THAT: In the discussion: you might like to look at the Carer Support Needs Assessment Tool (CSNAT) Approach at http://csnat.org/ - the CSNAT includes an item about “knowing what to expect in the future” which could help start a supportive conversation with a health or social care professional about this topic. YOU RESPONDED SAYING: Thank you for this suggestion. We had looked at it previously and although this is a great tool, we do not think it really addresses the carers specific concerns about care of their care recipient if they are not able to do so. We have therefore not cited it here. MY RESPONSE BACK: This is not how CSNAT works - it deliberately doesn't include lots of very specific questions about different support needs (such as worrying about what might happen when they can no longer care) as it would be way too long and unusable in clinical practice. Instead it includes 14 broad areas of support need, one of which relates to "Knowing what to expect in the future" which a carer might tick because they are worried what will happen in the future when they can no longer care. This could start a conversation with a clinician about this worry/need - this is how CSNAT works. I would urge you to reconsider mentioning it in the discussion. I should add that I am NOT an author of the CSNAT - I am just recommending it as a useful potential evidence-based intervention (and there are very few of those for carers) that might identify this worry/concern in carers.
---

VERSION 2 – AUTHOR RESPONSE

Thank you for these comments.

We have clarified the Editor's point by adding 'with pseudonyms' (highlighted in red font).

Page 9 now reads 'quotes are anonymised with pseudonyms'.

We thank Reviewer 2 for the final comment and have reflected on it at length but after deliberation have decided not to incorporate the CSNAT reference. This is a tool used when the person being cared for is at the end of their life. It therefore is not appropriate to use it with carers before the care recipient is coming to the end of their life - indeed it might be distressing.